# Motivation for and Effect of Cooking Class Participation: A Cross-Sectional Study Following the 2011 Great East Japan Earthquake and Tsunami

**DOI:** 10.3390/ijerph17217869

**Published:** 2020-10-27

**Authors:** Ai Tashiro, Kayako Sakisaka, Yuri Kinoshita, Kanako Sato, Sakiko Hamanaka, Yoshiharu Fukuda

**Affiliations:** 1Graduate School of Environmental Studies, Tohoku University, 468-1, Aoba, Aramaki, Aoba-ku, Sendai, Miyagi 980-8572, Japan; 2Teikyo University Graduate School of Public Health, 2-11-1 Kaga, Itabashi-ku, Tokyo 173-8605, Japan; sakiko.hama@gmail.com (S.H.); fukuday@med.teikyo-u.ac.jp (Y.F.); 3Division of Food Science and Nutrition, Tohoku Seikatsu Bunka Junior College, 1-18-2, Niji-No-Oka, Izumi-Ku, Sendai, Miyagi 981-8585, Japan; y.kinoshita@mishima.ac.jp; 4Department of Health and Nutrition, Junior College Course, Chukyo Gakuin University, 2216, Toki-Tyou, Mizunami, Gifu 509-6192, Japan; k-sato@chukyogakuin-u.ac.jp

**Keywords:** cooking class, post-disaster support, eating behavior, solo dining, health promotion, regional study

## Abstract

We explored the association between the motivation for and effects of cooking class participation in disaster-affected areas following the 2011 Great East Japan Earthquake and Tsunami. We conducted questionnaire surveys in January and February 2020, and applied three Poisson regression models to a cross-sectional dataset of participants, analyzing three perceived participation effects: increase in new acquaintances and friends, increase in excursion opportunities, potential for gaining motivation, and a new sense of life purpose. We also applied the interaction term of motivation variables and usual eating patterns (eating alone or with others). We obtained 257 valid responses from 15 cooking venues. The interaction term for participants’ motivation and eating patterns was associated with their perceived participation effects. “Motivation for nutrition improvement × eating alone” was positively associated with an increase in new acquaintances and friends (IRR: 3.05, 95% CI, 1.22–7.64). “Motivation for increasing personal cooking repertoire × eating alone” was positively associated with increased excursion opportunities (IRR: 5.46, 95% CI, 1.41–21.20). In contrast, the interaction effect of “motivation of increasing nutrition improvement × eating alone” was negatively associated with increased excursion opportunities (IRR: 0.27, 95% CI, 0.12–0.69). The results show that the cooking class was effective, as residents’ participation improved their nutritional health support and increased their social relationships.

## 1. Introduction

Natural disasters affect some eating patterns and behavior among affected residents [1]. Due to lifestyle changes following a natural disaster, food and nutrition management becomes challenging and affects people’s eating patterns, depending on the post-disaster phase. Japanese cuisine (*washoku*) has come to be perceived internationally as healthy due to its use of fresh ingredients, simple cooking style, and well-balanced nutrition [2]. However, some researchers have suggested that the choices and methods of Japanese cuisine were limited after the disaster, and evacuees tended to eat unbalanced meals such as non-perishable foods high in carbohydrates.

Another aspect of such changes is that dietary options also tend to converge among residents with close social connections [3]. For the elderly, in particular, the nutrition of individuals who eat alone tends to be biased when compared with those who eat with families due to the difficulties of eating management. Some studies indicate that eating alone incurs negative impacts linked to numerous mental and physical health conditions, such as depression, diabetes, and high blood pressure [4]. According to an agriculture ministry study, the proportion of people over 60 who occasionally eat alone was 23% for men and 28% for women in 2018 [5].

Those who dine alone almost daily account for nearly 25% of women over 70. Living and eating alone can cause various issues, especially for the elderly, such as health deterioration (e.g., diabetes, obesity, and hyperlipidemia) and extreme loneliness, attributed to a lack of communication [5]. Moreover, eating behavior is strongly influenced by a social context [6]. As mentioned above, dietary choices converge among individuals within close social connections [6]. Thus, to manage and support the dietary behavior and nutrition of disaster-affected people, establishing a diet and nutrition system to secure the necessary food, and supporting disaster survivors by establishing nutritious dietary habits are vital. However, such a social eating support system is yet to be established. Therefore, its potential in responding to post-disaster changes in eating patterns and behaviors has not been examined [7].

After the 2011 Great East Japan Earthquake and Tsunami (GEJET), volunteers providing food aid and support served hot meals to disaster victims [8]. Such volunteer activity is regarded as a practical social eating support. A prior study suggested that individuals with low self–control are likely to follow perceived peer eating norms [9]. Social eating norms constitute a novel target for interventions to encourage healthier eating [10,11,12]. However, those responses were temporary, focusing on the immediate post-disaster phase [13]. Appropriate eating norms are rooted in local cultural contexts and established through the behavior of residents. However, volunteers and aid organizations rarely could not consider such local customs, norms or food traditions when providing disaster-affected areas with immediate nourishment.

Following the initial GEJET response, the Red Apron Project was established in October 2011, as part of the Ajinomoto Group’s ongoing initiative to support reconstruction in the Tohoku region (Iwate, Miyagi, and Fukushima Prefecture) [14]. The cooking classes delivered through this initiative was aimed at improving residents’ nutrition, ascribing to the motto “Eat Well, Live Well” [14,15]. Enlisting voluntary participation by Ajinomoto Group staff, the project provides local participants with opportunities to learn about food and cooking—a rare type of post-disaster nutrition intervention. The cooking classes include social contribution activities to support reconstruction through nutritional improvement [15].

The Ajinomoto Foundation (TAF) successfully implemented cooking class activities in 2017. The concept of TAF is to act as a moderator between local communities, regional governments, and non–profit organizations in disaster-stricken areas, managed by dedicated project supervisors [15]. TAF also promotes the sustainability of social businesses and activities. The region’s food and nutrition issues were revealed in interviews with officials from local governments, social welfare councils, universities, and non profit organizations (NPOs). 

In the post-GEJET context, one emerging issue was the emerging health problems caused by unbalanced diets, partially due to decreased cooking activities in the small kitchens of temporary housing. When the elderly joined temporary housing communities, social interactions between residents were also weak and many tended to be withdrawn and feel isolated [15]. To address unbalanced dietary behavior and social issues in the disaster affected areas, the Ajinomoto Group started their cooking class activities in the Tohoku region based on the company’s philosophy, “Ajinomoto Group Shared Value,” which aimed to solve local social issues through the group’s business, and to share the cooking methods with local people, and to improve regional economic growth [16]. The Ajinomoto Group participated in the United Nations Global Compact (UNGC) in July 2009, and the activities of TAF were designated a core issue of the Sustainable Development Goals (SDGs) adopted at the United Nations General Assembly in September 2015 [16]. It is difficult for companies to independently resolve social isolation issues among disaster-affected communities, emphasizing the importance of developing solutions in cooperation with local communities and stakeholders.

Decades of studies have produced well-documented evidence from food diaries, revealing the social facilitation of eating [17,18,19]. However, the intentions of volunteers and aids do not always match the needs of beneficiaries [20]. Limited studies have examined the kinds of post-disaster dining activities or eating supports that have motivated the survivors in the past, including the results of current eating support interventions [21,22,23]. Furthermore, there is currently no existing research on the effectiveness of a free cooking class to train, encourage, and motivate disaster victims negatively affected nutritionally after a crisis.

This study aimed to explore the effects of cooking classes facilitated by the Ajinomoto Group (2011–2016) and TAF (2017–2020) on survivors of the GEJET, and to identify what kinds of motivations are associated with perceived effects. Furthermore, the study examined whether the cooking classes, as participatory health and nutrition promotion interventions, were successful in terms of health promotion and as a tool to assist residents in rebuilding social relationships.

## 2. Materials and Methods

### 2.1. Aim, Design, and Setting of the Study

This study aimed to explore the impact of the cooking support (participatory health and nutrition lectures), hosted by the Ajinomoto Group and TAF, on survivors of the GEJET living in disaster-affected areas. The voluntary cooking class was launched as an intervention in October 2011. The Ajinomoto Group dispatched employees who volunteered as cooking staff to the disaster–affected areas of Iwate, Miyagi, and Fukushima Prefecture (Figure 1). The class was aimed at resolving social and nutritional issues in these communities after the disaster by addressing cooking and eating behavior.

Until February 2020, more than 4100 cooking classes had been facilitated, and around 63,000 participants joined the classes [24]. The class is a fully free event. Ajinomoto staff, together with local stakeholders, determined the venues and menus, while the staff procured and prepared the ingredients prior to each class. Local stakeholders recruited participants, who were mostly elderly. A separate cooking class specifically for local men was also held regularly.

Excessive salt intake causes high blood pressure, leading to the onset of various lifestyle-related diseases [25,26,27]. Thus, the Ajinomoto Group started spreading information on low and appropriate salt intake in the Tohoku region, Japan, from 2014. The company continues to offer menu suggestions featuring local ingredients, conducts cooking classes for nutritionists and members of local nutrition improvement councils, and engages in additional activities to raise awareness of social and nutritional issues in the Tohoku region. Recipes aim to deliver around 500 kcal per meal, containing a maximum of 3 g of salt, and at least 20 g of protein. Dishes are easy to cook and use readily available, low-cost, and seasonal ingredients (less than 300 yen per meal). At the start of each cooking class, a staff member presents a brief nutrition lecture using a form of storytelling called *kamishibai* in Japanese (literally, “paper drama”) (Figure 2). Following these guidelines, the lectures help participants to implement the cooking process more easily.

### 2.2. Participants

To recruit participants, the Ajinomoto Group and TAF approached local governments and public institutions (mainly social welfare councils), councils for improving eating habits, and temporary housing residents’ associations. The head of each organization announced the cooking classes to the residents on a community circular board (*kairanban*). The number of participants was not specified or limited. However, most potential participants were elderly people who could participate during the daytime on weekdays.

This study utilized a questionnaire survey after each cooking class. Overall, 260 participants joined group cooking classes and classes for men in 15 venues between January and February 2020. The participants included evacuees who lived in disaster public housing. Some participants moved from other locations following the GEJET. In Fukushima Prefecture, most participants had relocated due to the radiation influence of the Fukushima Daiichi nuclear accident in 2011. The administered questionnaire included questions on participants’ personal characteristics, disaster experience and related damages, outcomes of their cooking class participation, motivation for their cooking class participation, and changes in eating patterns and dietary behaviors before and after the GEJET.

Participants provided written informed consent before completing the questionnaire survey. Data obtained from participants’ questionnaire responses were anonymized. Ethical approval was obtained from Teikyo University Ethical Review Board for Medical and Health Research involving Human Subjects (Approved No.19-248, 16 January 2020).

### 2.3. Analysis

#### 2.3.1. Dependent Variables

Three effect measurements of cooking class participation, labeled E1 to E3, were used as dependent variables.

E1: increase in new acquaintances and friendsE2: increase of excursion opportunitiesE3: gaining a new sense of life purpose

Likert response options included: 1 = “Strongly agree,” 2 = “Agree somewhat,” 3 = “Disagree somewhat,” and 4 = “Strongly disagree.” We categorized the responses as follows: “Strongly agree” and “Agree somewhat” were designated a positive effect (value = 1), while “Disagree somewhat” and “Strongly disagree” were designated a negative effect (value = 0).

#### 2.3.2. Independent Variables and Covariates

The primary independent variables of interest were participants’ motivations, labeled M1 to M6. They were measured by the question “Why did you decide to join the cooking class?” in the following categories.

M1: I am interested in improving nutrition.M2: I wanted to learn about healthy cooking.M3: I wanted to increase my cooking repertoire.M4: I wanted to eat with someone.M5: I wanted to make new acquaintances and friends.M6: I wanted to receive a souvenir after the cooking class.

Potential answers were 1 = “Strongly agree,” 2 = “Agree somewhat,” 3 = “Disagree somewhat,” and 4 = “Strongly disagree” for each option, using a Likert scale. We categorized the responses as follows: “Strongly agree” and “Agree somewhat” were assigned a value of 1, while “Disagree somewhat” and “Strongly disagree” were assigned a value of 0.

Gender, age, occupation, number of family members, current economic status compared to pre-GEJET status, duration (years) of living in the current house, housing type, types of disaster damage, cooking independently, eating independently, poor mental health, self-rated health, coastal or inland residence, frequency of cooking class attendance, and venue were the covariates. We categorized participants’ age into three ranges: “20 to 64,” “65 to 74,” and “75 or over.” The employment status was categorized as “employed” and “unemployed.” Family size (number of family members living in the household, including the respondent) was categorized as “1,” “2–3,” and “4 or more.” Compared to their pre-GEJET budget, the family budget status was considered as good or bad economic status based on the participants’ response (“Severe,” “somewhat severe,” “neither,” “somewhat better,” and “better”). “Severe” or “somewhat severe” were categorized as bad economic status (value = 1) and “somewhat better” or “better” were categorized with a good economic status (value = 0). Duration of living at the current place was categorized as “less than 1 year,” “less than 5 years,” “less than 10 years,” “less than 20 years,” and “20 years or more.” Type of housing was categorized as “own housing,” “rental or temporary housing, housing of acquaintance/family/relatives, or others,” and “Disaster public housing.” Types of disaster damage were “tsunami,” “fire,” “Fukushima Daiichi Nuclear Power Plant Accident,” “reputational damage,” “no damage,” “placed in non-disaster affected areas,” and “others.”

Regarding health status, we applied self-rated health (SRH) [28] and psychological distress measures, using the Kessler Psychological Distress Scale (K6) [29]. SRH is a simple, single-item question about participants’ health status, and is considered a valid measurement of health in epidemiological surveys. The participants were asked, “What is your physical health condition today?” The potential answers were “very good,” “good,” “not so good,” and “bad”. The first two responses indicated good SRH and the last two indicated poor SRH. K6 was originally developed to screen for non-specific psychological distress in mental health research. Its credibility and availability were documented in previous studies [29]. This study used the Japanese version of the K6 questionnaire, which has been validated [30]. K6 consists of a six-item battery exam, asking how frequently respondents have experienced symptoms of psychological distress in the past 30 days. The responses range from “0: none” to “4: all of the time,” with a total score ranging from 0 to 24. Following previous studies [30], this study categorized the total score into categories of 13 or more (severe psychological distress) and 12 or under (no severe psychological distress).

Residence at the time of the GEJET was categorized as “inland” and “coast” based on whether it was in a tsunami-affected area or not. The cooking class venues were used as control variables because the venues were randomized, and the effects of participation may have differed depending upon the regional contexts.

### 2.4. Statistical Analysis

This study calculated the incidence rate ratios (IRR) with 95% confidence intervals (CIs) for the effects of cooking class participation by the exponents of coefficients of the fitted Poisson regression model. Considering an overestimation problem, this study replaced IRR with the odds ratio for the binary outcome because the odds ratio should be calculated when the incidence of the outcome is less than 10% [31]. In each model, this study interacted the eating alone dummy with the main interest variables (six motivations of participation) to calculate whether there was a difference in motivation for participation based on whether the participant ate alone.

Furthermore, this study computed the marginal effects of motivation for eating alone or not to evaluate the difference in the impact on participation effects (dy/dx). Data were analyzed using Stata version 15.0 (StataCorp LP, College Station, TX, USA), and the level of statistical significance (*p*-value) was set at 0.05.

## 3. Results

### 3.1. Participants’ Basic Characteristics

After obtaining data from 260 participants, we excluded missing data for the three perceived effects, ultimately including valid data from 257 participants. Table 1 shows the characteristics of participants: 79.77% were women, the greatest ratio of participants’ age was 65–74 years old (46.09%), approximately 80% were aged 65 or older, 83.96% were unemployed, and 15.95% lived alone.

Of the participants, 49.03% had lived in their current living space for 20 years or more. Among the types of damage related to the GEJET, 45.33% of participants claimed damages from the tsunami, and 14.79% of participants indicated no damage. Regarding the preparation of meals, the percentage of participants cooking by themselves was 23.35%. The percentage of participants eating alone was 17.90%. Regarding health status, 36.19% of participants claimed poor SRH, and 3.5% claimed poor mental health (k6 ≥ 13). The average number of cooking classes attended was 7.23 (± 9.95).

### 3.2. Motivation for and Effects of Cooking Class Participation

Table 2 shows the results of the association between motivation to participate in the cooking class and the perceived effects of participation, based on Poisson regression analysis. In model 1, all motivations for making new acquaintances and friends (M5) were positively associated with the effect of increased new acquaintances and friends after the cooking class (E1) (IRR: 1.45, 95% CI, 1.12–7.64). Eating patterns were not associated with E1 (*p* > 0.05), even though the interaction effects of motivation and eating patterns were. “Motivation for nutrition improvement (M1)× eating alone” was positively associated with E1 (IRR: 3.05; 95% CI, 1.22–7.64). In contrast, “motivation for increasing cooking repertoire (M3) × eating alone” was negatively associated with E1 (IRR: 0.33; 95% CI, 0.17–0.67). The difference of E1 between venues was significant except in venue i (*p* < 0.001).

Model 1 to 3 included cooking class venues, motivations, eating patterns, interaction effects of motivation, and eating patterns with adjustment for age, gender, occupation, number of family members, economic status compared with pre–GEJET, number of years living at their current domicile, housing type, types of disaster damage, cooking by oneself, eating alone, poor mental health, self–rated health, coastal or inland housing, number of the cooking classes attended, and venue dummy (Appendix A, Appendix A).

In model 1, all motivations for making new acquaintances and friends (M5) were positively associated with the effect of increased new acquaintances and friends after the cooking class (E1) (IRR: 1.45, 95% CI, 1.12–7.64). Eating patterns were not associated with E1 (*p* > 0.05), but the interaction effects of motivation and eating patterns were associated with it. “Motivation for nutrition improvement (M1)× eating alone” was positively associated with E1 (IRR: 3.05; 95% CI, 1.22–7.64). In contrast, “motivation for increasing cooking repertoire (M3) × eating alone” was negatively associated (IRR: 0.33, 95% CI, 0.17–0.67). The difference of E1 among venues were significant except in venue *i* (*p* < 0.001).

In model 2, M1 was positively associated with the effect of increased excursion opportunities (E2) (IRR: 2.43, 95% CI, 1.55–3.81). Regarding interaction effects, M1 and M5 for people eating alone were negatively associated with E2 (M1× eating alone: IRR: 0.27, 95% CI, 0.08–0.90, M5× eating alone: IRR: 0.37, 95% CI, 0.18–0.78). In contrast, M3 × eating alone was positively associated with E2 (IRR: 5.46, 95% CI, 1.41–21.20). Regarding venues, venue f was negatively associated with E2 (IRR: 0.39, 95% CI, 0.20–0.76).

In model 3, M1 and M5 were positively associated with the effect of gaining a new sense of purpose in life (E3)( IRR: 1.90, 95% CI, 1.01–3.59, IRR: 1.47; 95% CI, 1.12–1.93, respectively). Regarding the interaction effects, M1 × eating alone was negatively associated with E3 (IRR: 0.28, 95% CI, 0.12–0.69). There were no significant differences in venues (*p* > 0.05).

Figure 3 shows the results of the estimated marginal effects of eating patterns on participation effects, according to motivations. The horizontal axis indicates the mean increase in the predicted probability of participation effect (dy/dx) of E1 to E3 by eating patterns (eating alone or not) based on the fitted Poisson regression model.

Regarding dy/dx in model 1, E1 of people eating alone with M1 and M5 was significantly higher than those without M1 or M5 (dy/dx: 0.56, *p* = 0.003, dy/dx: 0.46, *p* = 0.006, respectively). E1 of people eating alone with M3 was lower than those without 3 (dy/dx: −1.25; *p* = 0.037). In model 2, E2 of people eating alone with M3 and M4 was higher than those of people eating with someone. E2 of people eating alone with M3 and M4 was significantly higher than without M3 and M4 (dy/dx: 0.58, *p* < 0.001, dy/dx: 0.49, *p* = 0.002, respectively). E2 of people eating with someone with M5 was higher than that of people eating alone with M5 (dy/dx: 0.42, *p* < 0.001, dy/dx: −0.59, *p* = 0.76, respectively). In models 3 and 4, change of dy/dx of eating alone and with someone indicated the same trends. However, dy/dx of people eating alone showed no significant changes in model 3 (*p* > 0.05). In contrast, dy/dx of people eating with another individual with M1 and M5 was higher than those without M1 and M5 (M1: dy/dx: 0.34, *p* = 0.007, M5: dy/dx: 0.24, *p* = 0.001).

## 4. Discussion

This study explored the impact of cooking classes hosted by TAF on residents who were negatively affected by the GEJET, and identified the kind of motivations associated with participants’ perceived effects of the cooking class. The results showed that motivation and participation effects differed, depending upon whether participants eat alone or not. This study revealed that the motivation for increasing one’s cooking repertoire was positively associated with achieving a balance between work and private life. Motivation to eat with someone was positively associated with getting involved in the management of cooking classes. The motivation for making new acquaintances and friends was positively associated with excursion opportunities and achieved a balance between work and private life.

Some results can be explained as participants seek reconnection with local people and realize that social cohesion could be established by attending the cooking classes. Those results indicate that an opportunity to interact with residents again existed. As a result, grouping behavior worked as a trigger, shaping new neighborhood relationships. In contrast, the motivation for making new acquaintances and friends was negatively associated with becoming involved in the management of cooking classes. This association seems to indicate that participants were only seeking horizontal connections but did not intend to manage such cooking classes or demonstrate leadership as a participation effect.

Regarding the interaction effects between motivation and eating alone, the motivation of nutrition improvement was positively associated by increasing new acquaintances and friends and becoming involved in the management of cooking classes. The motivations of increasing their cooking repertoire and people eating alone was positively associated with excursion opportunities. IRR of the interaction was the largest value as the interaction effect in model 2 (IRR: 5.46, 95% CI, 1.41–21.2). This association can be explained by residents’ frustration in planning well–balanced meals after the disasters. However, participants increased their cooking repertoire through participation in the cooking classes, identified more excursion opportunities, and could teach other people or deliver meals to their acquaintances and friends. In addition, cooking has been shown to be effective in preventing dementia [32], as it involves the execution of a series of tasks such as preparation, cooking, and seasoning, which stimulates the frontal lobe [32]. Teaching cooking not only increases interaction between residents but also strengthens participants’ minds and activates their brain [33,34].

The motivation for making new acquaintances and friends among people eating alone was negatively associated with excursion opportunities. This can be explained by comparing people who eat with someone and people eating alone with their intention of making friends, which was not reflected in the increase of excursion opportunities through cooking participation. In Fukushima Prefecture, several people had relocated. Therefore, it took time to acquaint themselves with other residents who had lived there before the GEJET [35]. Challenges in building new relationships within new neighborhoods were also revealed [36]. A difference in the cultures between the inland and coastal areas also influenced social connections [37].

Moreover, in disaster-affected areas, the isolation of the elderly living alone has become a problem. So far, empirical studies suggest that social support is an effective way to improve their quality of life [38,39]. Studies suggest the effectiveness of social support [37,38], but no specific support system has been presented [40,41,42]. However, at the practical level, taking out the withdrawn or isolated elderly is difficult [43].

Given the above negative issues in previous studies [38,39,40,41,42,43], this study presented a new support measure for survivors in disaster-affected areas by hosting cooking classes to encourage the social participation of the elderly. Specifically, we considered TAF activities to be useful for building social connections among residents.

Currently, post-disaster volunteer support for cooking classes is rarely conducted [44]. There are few cases where cooking classes are used as tools to support long-term reconstruction volunteer activities, focusing on improving nutrition intake under local social norms [9]. After the disaster in the Tohoku region, there are fewer opportunities for local people to sit together at the dining table. Eating habits have grown more diverse because of trends toward moving away from nuclear families and lifestyle diversity. This tendency became remarkable after the GEJET in the Tohoku region, where no similar dietary improvement activities had been taking place. In such communities, people eating alone or eating personalized meals, even when together with family members, also rose. Moreover, with increasingly busy lives, people may not have the time to prepare healthy meals.

In contrast, this study identified that, in a community where dietary improvement support had been conducted, meals were for receiving nourishment, and local people communicated by sharing the joy of eating together and creating more free time through smart cooking that involves enjoyable and efficient meal preparation. In this sense, the cooking class is useful because Japanese people appreciate an opportunity to cook since their childhood. In Japan, cooking is part of the elementary school curriculum.

Through cooking and eating with people, conversations occur naturally. If the local government provides cooking classes, the local community’s social capital will be enhanced [41]. For instance, employing the regional social norms, such as sharing a similar menu with other participants and grandchildren who live far away and eating together virtually will make the elderly feel closer to them, increase their happiness, and stimulate further conversations [12]. Although this initiative is a company’s voluntary activity, presented at no cost to participants, TAF has committed to assist in facilitating the future independent operation of the class by local residents. Further, a few years after the inception of the TAF intervention, some local governments have started facilitating independent cooking class activities at the cost of about $ 4.5 to participants.

Few previous studies have shown evidence of the long–term effects of food support volunteer activities [45,46]. According to the company’s survey, people who regularly prepare meals, prioritize nutritionally-balanced menus, while 76% of people feel stressed by planning menus with multiple courses. To address this issue, the company developed a system that instantly suggests two suitable items from among soup, an entree, and a side dish. Using Artificial Intelligence (AI), the database advises meal preparers regarding nutrition calculation, seasonality, cuisine type (Japanese, Western, or Chinese), and efficiency of meal preparation (number of ingredients and cooking utensils) that consider unquantifiable and inexpressible sensory elements, such as color and taste balance. Participants use the recipe and cook by themselves after a cooking class by accessing the database on Ajinomoto’s website [15]. The method also improves residents’ mental health (promoting social participation) and physical health (taking nutritional diets). In this sense, this study is the first that quantified the effect of corporate volunteer supports in the disaster areas.

There are several limitations to this study. First, there is a bias in the location of the survey and the number of participants. This study was conducted only with program participants, with no control group comprising people participating in other events, or people who had not attended any other event. A large group of residents did not participate, and this study could not determine their reasoning for not participating in the class. Thus, to prevent future bias, controlled trials should have been conducted. Second, there is a gender bias of respondents. Therefore, robust data could not be used. Additionally, this is a cross-sectional study, where an estimation of change of effect and motivation from a cohort study is required. Future studies also need to investigate the effects of program design on the effectiveness of cooking classes, with consideration given to aspects such as the ideal number of exposures, changes in cooking interventions, and the usefulness of combined activities. However, this study does not believe that this limitation affected our results because the average number of attendances regarding the cooking classes was 7.5 times. This fact demonstrates that the data used in this study included people who have continued to participate in the cooking class. Therefore, the data included the effect of intensity or frequency of social participation suggested in previous studies [47,48].

## 5. Conclusions

The motivation for participating in cooking classes was associated with the effects of participants’ social aspects and dietary behaviors. Individuals eating alone with the motivation of improving nutrition made more new acquaintances and friends, and they were more motivated to become involved in managing the program than individuals who had been eating with others. People eating alone with the motivation to improve nutrition had a greater desire to increase excursion opportunities than people eating with someone.

Previous studies focused on post-disaster food supply as a matter of physical survival in an emergency. By contrast, this study takes long-term post-disaster nutrition health to the next level, evaluating the balance of a diet and considering and prioritizing cultural dietary aspects that embrace traditional heritage and history. Such aspects encourage the re-establishment of socialization and fellowship around a table as well as improvement in mental health and emotional recovery from the physical and emotional toll such crises can have. Establishing a cooking class as a post-disaster rehabilitation for social cohesion is a new method to break through previous post-disaster social participation theories [43,47,48]. The most beneficial and practical aspect of this study is the social dining methodologies designed to embrace local culture and cuisine to create a sense of normalcy in a post-disaster context. The social dining methodologies also encourage fellowship and motivate individuals to advance beyond their current dire circumstances.

## Figures and Tables

**Figure 1 ijerph-17-07869-f001:**
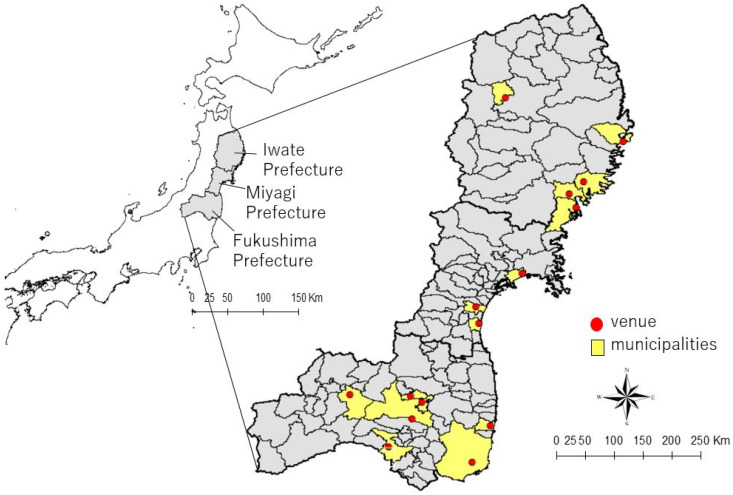
Study areas where cooking classes were held.

**Figure 2 ijerph-17-07869-f002:**
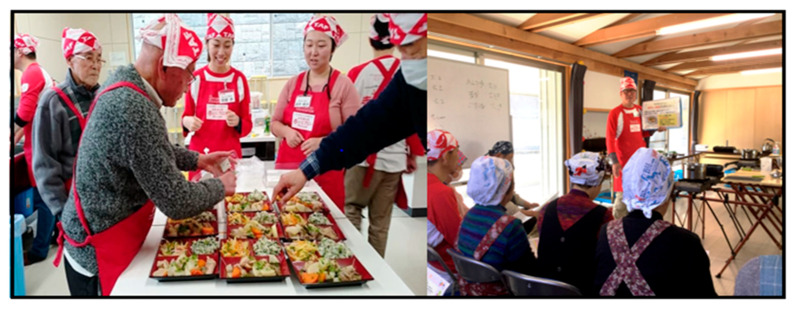
Cooking class: (Left) Participants’ cooking activity. (Right) Lecture using *Kamishibai.* (photos by authors).

**Figure 3 ijerph-17-07869-f003:**
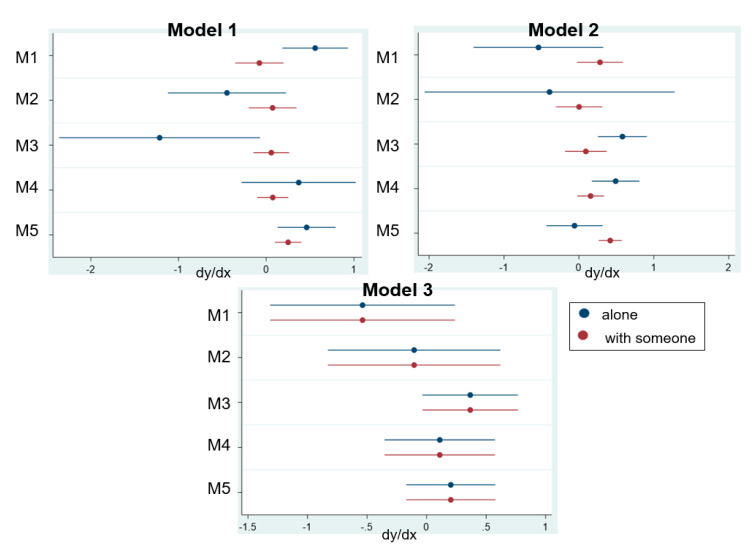
Marginal effects of eating pattern on participation effects by motivation.dy/dx is the discrete change in participation effects of margins by motivation with an eating pattern based on the fitted Poisson regression model in Table 2 (Appendix A, Appendix A).

**Table 1 ijerph-17-07869-t001:** Characteristics of participants.

	N = 257	%
Dependent Variables: Participation Effects		
E1: Increase in New Acquaintances and Friends		
yes	192	74.71
no	65	25.29
missing	0	0.00
total	257	100.00
E2: Increase in excursion opportunities		
yes	151	58.75
no	106	41.25
missing	0	0.00
total	257	100.00
E3: Gaining a new sense of life purpose		
yes	176	68.48
no	81	31.52
missing	0	0.00
total	257	100.00
**Independent Variables: Motivations for Participation**		
M1: Nutrition improvement		
yes	232	92.07
no	20	7.78
missing	5	1.95
M2: Learn healthy cooking		
yes	227	88.33
no	24	9.34
missing	6	2.33
total	257	100.00
M3: Increased cooking repertoire		
yes	217	84.44
no	34	13.23
missing	6	2.33
total	257	100.00
M4: Eat with someone		
yes	194	75.49
no	58	22.57
missing	5	1.95
total	257	100.00
M5: Make new acquaintances and friends		
yes	171	66.54
no	81	31.52
missing	5	1.95
total	257	100.00
M6: Receive a souvenir		
yes	52	20.23
no	197	76.65
missing	8	3.11
total	257	100.00
Cooking class venues		
a	16	6.23
b	18	7.00
c	33	12.84
d	7	2.72
e	13	5.06
f	31	12.06
g	16	6.23
h	24	9.34
i	11	4.28
j	17	6.61
k	13	5.06
l	10	3.89
m	16	6.23
n	16	6.23
o	16	6.23
missing	0	0.00
total	257	100.00
**Covariates**		
Personal characteristics		
Eating alone		
yes	46	17.90
no	209	81.32
missing	2	0.78
total	257	100.00
Age, year		
20–64	52	20.31
65–74	118	46.09
75 or more	86	33.59
missing	1	0.01
total	257	100.00
Occupation		
employed	41	16.02
unemployed	215	83.96
missing	1	0.02
total	257	100.00
Family size		
1	41	15.95
2	111	43.19
3	40	15.56
4 or more	65	25.29
missing	0	0.00
total	257	100.00
Economic status		
bad	83	32.30
good	172	66.93
missing	2	0.78
total	257	100.00
Number of years in the current place		
< 1 year	12	4.67
1–4 years	61	23.74
5–9 years	53	20.62
10–19 years	5	1.95
20 years ≤	126	49.03
missing	0	0.00
total	257	100.00
Current housing type		
own	195	75.88
rented housing or temporary housing, housing of acquaintance/family/relatives, or others	46	17.9
disaster public housing	15	5.84
missing	1	0.39
total	257	100.00
Disaster damage (multiple answered allowed)		
tsunami	117	45.33
fire	6	2.33
Fukushima Daiichi Nuclear Power Plant Accident	80	31.13
no damage	38	14.79
placed in non-disaster-affected areas	29	11.28
others	32	12.45
missing	0	0.00
total	257	100.00
Cooking by oneself		
yes	197	23.35
no	60	76.65
missing	0	0.00
total	257	100.00
Self–rated health		
poor	93	36.19
good	163	63.42
missing	1	0.39
total	257	100.00
Mental health		
poor	9	3.50
good	243	94.55
missing	5	1.95
total	257	100.00
Number of the cooking classes attended †	7.23	9.95
Area		
coastal	136	52.92
inland	121	47.08
missing	0	0.00
total	257	100.00

† mean ± SD.

**Table 2 ijerph-17-07869-t002:** The association between participants’ motivation and their perceived effects.

	Model 1: E1		Model 2: E2		Model 3: E3	
	IRR	95% CI	IRR	95% CI	IRR	95% CI
**Independent variables**						
**Motivation**						
M1: Nutrition improvement						
	0.90	(0.64, 1.27)	1.87	(0.74, 4.74)	1.90 *	(1.01, 3.59)
M2: Learn healthy cooking						
	1.11	(0.74, 1.66)	1.00	(0.59, 1.70)	1.03	(0.67, 1.60)
M3: Increase cooking repertoire						
	1.08	(0.81, 1.45)	1.18	(0.69, 2.03)	0.95	(0.68, 1.33)
M4: Eat with someone						
	1.11	(0.86, 1.43)	1.34	(0.92, 1.94)	1.16	(0.87, 1.55)
M5: Make new acquaintances and friends						
	1.45 ***	(1.12, 1.86)	2.43 ***	(1.55, 3.81)	1.47 **	(1.12, 1.93)
M6: Receive a souvenir						
	Reference		Reference		Reference	
**Eating patterns**						
alone	0.83	(0.28, 2.49)	1.11	(0.16, 7.63)	1.84	(0.55, 6.20)
with someone	Reference		Reference		Reference	
**Interaction effect (motivation × eating pattern)**						
M1 × alone	3.05 *	(1.22, 7.64)	0.27 *	(0.08, 0.90)	0.28 **	(0.12, 0.69)
M1 × someone	Reference		Reference		Reference	
M2 × alone	0.57	(0.29, 1.12)	0.59	(0.10, 3.65)	0.83	(0.29, 2.39)
M2 × someone	Reference		Reference		Reference	
M3 × alone	0.33 **	(0.17, 0.67)	5.46 *	(1.41, 21.20)	2.17	(0.78, 6.04)
M3 × someone	Reference		Reference		Reference	
M4 × alone	1.55	(0.50, 4.80)	2.43	(0.83, 7.12)	1.03	(0.44, 2.40)
M4 × someone	Reference		Reference		Reference	
M5 × alone	1.36	(0.74, 2.52)	0.37 **	(0.18, 0.78)	0.95	(0.45, 1.99)
M5 × someone	Reference		Reference		Reference	
**Venue of cooking class**						
a	Reference		Reference		Reference	
b	8.47 **	(2.07, 34.68)	0.47	(0.16, 1.32)	1.49	(0.57, 3.91)
c	6.05 **	(1.62, 22.63)	0.42	(0.17, 1.04)	1.09	(0.47, 2.52)
d	8.45 **	(1.85, 38.49)	0.50	(0.15, 1.68)	1.77	(0.60, 5.22)
e	7.32 **	(2.12, 25.32)	1.33	(0.69, 2.55)	1.86	(0.90, 3.86)
f	5.49 **	(1.58, 19.11)	0.39 **	(0.20, 0.76)	1.01	(0.50, 2.04)
g	6.11 **	(1.57, 23.72)	1.23	(0.70, 2.14)	1.93	(0.96, 3.87)
h	5.53 **	(1.57, 19.51)	0.79	(0.44, 1.40)	1.65	(0.87, 3.13)
i	3.75	(0.85, 16.68)	0.31	(0.03, 2.86)	1.36	(0.45, 4.12)
j	11.92 ***	(2.88, 49.32)	0.82	(0.30, 2.24)	2.00	(0.79, 5.06)
k	9.05 **	(2.03, 40.35)	0.60	(0.19, 1.87)	2.00	(0.79, 5.06)
l	5.26 *	(1.12, 24.69)	0.40	(0.11, 1.44)	1.30	(0.43, 3.87)
m	5.77 **	(1.56, 21.35)	0.99	(0.42, 2.31)	1.47	(0.61, 3.54)
n	10.18 **	(2.27, 45.69)	0.66	(0.20, 2.15)	1.94	(0.68, 5.56)
o	9.12 **	(1.97, 42.28)	0.63	(0.20, 1.96)	2.13	(0.77, 5.92)
AIC	520.97		482.01		520.24	

Omitted missing variable. ref., reference item. * *p* <0.05, ** *p* < 0.01, *** *p* < 0.001. IRR, incidence rate ratio, CI, confidence interval, AIC, Akaike’s information criterion. All models were adjusted by covariates in Table 1 (Appendix A, Appendix A).

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
