# Peer review of "Motivation for and Effect of Cooking Class Participation: A Cross-Sectional Study Following the 2011 Great East Japan Earthquake and Tsunami"

_ijerph, 2020, doi:10.3390/ijerph17217869_

Round 1

Reviewer 1 Report

Abstract :

  • The language in this section needs to be clearer. There are some word omissions. Some sentence structures could be better. For example, lines 17, 19-20, 25-30.
  1. Introduction
  • Page 1, line 42 – I think the word is “eat” not “take”
  • Page 2, line75 – seems a word is missing. Are the authors trying to say that ATF succeeded in implementing….
  • Page 3, line 98 – should read “effects on….”
  1. Materials and Methods

2.1    Aim, design, and setting of the study

  • Page 3, line 104 – should read “on survivors” not “for survivors”
  • Page 3, line 108 – should be past tense- “classes had been held…”
  • Page 3, lines 113-114 is unclear
  • Page 3, line 119- should the focus be on a cooking process that people enjoy or one that people can easily implement the cooking process?
  • Page 4 – A word in the label for the map seems to be cut off – e.g. the word “study”. The label for the figure seems incomplete
  • This section does not talk about how participants were recruited. Was there criteria to be met beyond being a survivor of the Tsunami?

2.2 Data Collection

  • Page 4, line 125 - The word “collection” is missing a “C”
  • Page 4, line 132 – Put a coma after the word “class”
  • Page 5, line 135-136 - This sentence is unclear. Needs to be restructured
  • Before getting to analysis, it would be good to describe the data collection instrument. Was it quantitative or qualitative? Did it have domains or sub-topics? Did the authors use a Likert scale- - what was the range? What do “E” and “M” in 2.3.1 and 2.3.2 stand for?
  • Page 5, line169 – delete “considered” at the end of the line
  • Page 5, line 170- delete “current” before the word “family”
  • Page 6, line 200-201 is unclear.
  1. Results
  • Page 6, line 210-211 is unclear
  • Page 9, line 216-217 – The beginning of the sentence should read “Of the participants…”
  • Page 9, line 219- qualify what you are saying – It should read, “The percentage of participants…”
  • Page 10, line 235-240 – the sentence is too long. Consider breaking into 2 or 3 so it is easy to understand.
  • Page 11, line 247 – Start the sentence with an article. It should read “The difference…”
  1. Discussion
  • Are these findings consistent with any studies previously conducted or vice versa? It would be good to indicate this
  • Page 12, line 316 – There are extra characters in the sentence that form no word
  • Page 12 line 317-328 – There seems to be some contradiction in what is being stated. In the preceding narrative, it looks like the classes yielded positive results for social cohesion/interaction, but in these lines the authors are saying volunteer supported cooking classes were low, no sitting together at dinning tables to eat etc. This piece needs to be presented more clearly.
  • Page 13, lines 346-353- I think this piece needs to be written in past tense.
  • Page 13, line 361-367 – These sentences need to be reworked and made clearer.

Author Response

Thank you very much for the thoughtful and constructive feedback you provided regarding our manuscript. We appreciate your taking the time to review our manuscript. Our responses to your comments are as follows:

Abstract :

·        The language in this section needs to be clearer. There are some word omissions. Some sentence structures could be better. For example, lines 17, 19-20, 25-30.

Thank you very much for your suggestion. We modified sentences as possible as we can in the abstract clearer. Sentences in L. 25-30 explained the main results of interaction effects. The name of varubaels can not be simplified any more so that you may feel the problem of readability, but we keep the term according to the original questionnaire

  1. Introduction

·        Page 1, line 42 – I think the word is “eat” not “take”

Thank you very much for your suggestion. We revised the word, “eat.”

·        Page 2, line75 – seems a word is missing. Are the authors trying to say that ATF succeeded in implementing

Thank you very much for your suggestion. We revised the sentence as follows:

The Ajinomoto Foundation (TAF) succeeded in implementing cooking class activities in 2017

·        Page 3, line 98 – should read “effects on….”

Thank you very much for your suggestion. We revised the sentence as follows:

Thus, this study aimed to explore the effects of cooking classes facilitated by the Ajinomoto Group (2011-2016) and TAF (2017-2020) on survivors of the GEJET, and to identify what kinds of motivations are associated with perceived effects.

  1. Materials and Methods

2.1    Aim, design, and setting of the study

Page 3, line 104 – should read “on survivors” not “for survivors”

We modified the sentence as follows:

This study aimed to explore the impact of the cooking support (participatory health and nutrition lectures), hosted by the Ajinomoto Group and TAF, on survivors of the GEJET living in disaster-affected areas.

Page 3, line 108 – should be past tense- “classes had been held…”

We modified the sentence as follows:

more than 4,100 cooking classes had been facilitated

Page 3, lines 113-114 is unclear

We added and modified the sentences as follows:

Excessive salt intake causes high blood pressure, leading to the onset of various lifestyle-related diseases [25-27]. Thus, the Ajinomoto Group started spreading information on low and appropriate salt intake in the Tohoku region, Japan, from 2014.

Page 3, line 119- should the focus be on a cooking process that people enjoy or one that people can easily implement the cooking process?

We modified the sentence as follows:

Following these guidelines, the lectures help participants to implement the cooking process easily.

·        Page 4 – A word in the label for the map seems to be cut off – e.g. the word “study”. The label for the figure seems incomplete

We modified the title of Figure 1 as follows:

Figure 1. Study areas where cooking classes were held.

This section does not talk about how participants were recruited. Was there criteria to be met beyond being a survivor of the tsunami?

Regarding the criteria of survivors, there is no clarified criteria or definitions. The most of the participants were disaster victims (affected by tsunami, earthquakes, or radiation problems).

The Ajinomoto Group (AG), and Ajinomoto Foundation(TAF) financed and offered cooking classes to support disaster victims, including these local host organizations (We called them ‘local counter partners’). However, we invited all community people who lived in the disaster-affected areas. Of course, TAF prioritized inviting people living in temporary housing or disaster recovery public housing in the first 3-4 years after GEJET 2011.

These cooking classes were implemented mainly through local host organizations in disaster-affected areas. The Ajinomoto Group announced community people through public networks, public papers, or through community health social workers.

Later, TAF invited not only people living in temporary housing but also asked all community people living in disaster-affected areas, since ‘difficult communication gap between living in the temporary housing and not living there’ were widely found as discussed in other countries (see  Lancet. 2019 Apr 27;393(10182):1697-1698; Bizri AR, Dada BA, Haschicho MH.Defamed relations: host community and refugees. doi: 10.1016/S0140-6736(18)33181-7. PMID: 31034375 ).

As your suggestion, we revised the parts as follows:

To recruit participants, the Ajinomoto Group and TAF approached local governments and public institutions (mainly social welfare councils), councils for improving eating habits, and temporary housing residents’ associations. The head of each organization announced the cooking classes to the residents on a community circular board (kairanban). The number of participants was not specified or limited; however, most potential participants were elderly people who could participate during the daytime on weekdays.

2.2 Data Collection

·        Page 4, line 125 - The word “collection” is missing a “C”

Thank you for your suggestion. We corrected the word.

·        Page 4, line 132 – Put a coma after the word “class”

According to your suggestion, we added a comma after class.

·        Page 5, line 135-136 - This sentence is unclear. Needs to be restructured

We modified the sentence as follows:

The administered questionnaire included questions on participants’ personal characteristics; disaster experience and related damages; outcomes of their cooking class participation; motivation for their cooking class participation; and changes in eating patterns and dietary behaviors before and after the GEJET.

·        Before getting to analysis, it would be good to describe the data collection instrument. Was it quantitative or qualitative? Did it have domains or sub-topics? Did the authors use a Likert scale- - what was the range? What do “E” and “M” in 2.3.1 and 2.3.2 stand for?

Data collection was explained in the first paragraph in section 2.2.

This analysis included both quantitative or qualitative variables with categorized items. You can see more about in section 2.3.1 and 2.3.2.

Three cooking classes’ participation effect measurements were applied as dependent variables:

  • E1: increase in new acquaintances and friends
  • E2: increase of outing opportunities
  • E3: gaining a new sense of purpose in life

Response options included 1 = “Strongly agree,” 2 = “Agree somewhat,” 3 = “Disagree somewhat,” and 4 = “Strongly disagree.” The study categorized the responses as follows: “Strongly agree” and “Agree somewhat” were changed to a positive effect (value = 1) and “Disagree somewhat” and “Strongly disagree”were changed as a negative effect (value = 0).

2.3.2. Independent variables and covariates

The primary independent variables of interest were participants’ motivations. They were measured according to the following questions:

“Why did you decide to join the cooking class?”

M1) I am interested in improving nutrition.

M2) I wanted to learn about healthy cooking.

M3) I wanted to increase my cooking repertoire.

M4) I wanted to eat with someone.

M5) I wanted to make new acquaintances and friends.

M6) I wanted to receive a souvenir after the cooking class.

Potential answers were, 1 = “Strongly agree,” 2 = “Agree somewhat,” 3 = “Disagree somewhat,” and 4 = “Strongly disagree” for each option. We categorized the responses as follows: “Strongly agree” and “Agree somewhat” were changed the value into 1 and “Disagree somewhat” and “Strongly disagree” were changed the value into 0.

E indicates the effects of three questions and M indicates the motivations of 6 questions. Each name is long, so we headed them for simplified names like E1, E2, E3, M1, M2, M3, m4, M5, M6.

·        Page 5, line169 – delete “considered” at the end of the line

We deleted the word.

·        Page 5, line 170- delete “current” before the word “family”

We deleted the word.

·        Page 6, line 200-201 is unclear.

Regarding this sentence, we explaned the over estimation problem. In general, when an outcome is presented as 0 and 1, odds ratio by logit model is calculated. However, as the refenrece paper [31] mentioned, when the incidence of the outcome variable is over 10%, odds ratio is not recommended. Thus, we applied poisson regression model and calculated IRR in this study.

We modified this sentence as follows:

Considering an overestimation problem, this study replaced IRR with the odds ratio for the binary outcome, because the odds ratio should be calculated when the incidence of the outcome is less than 10%.

  1. Results

·        Page 6, line 210-211 is unclear

This sentence means the response number of participants was 260. Out of these, 3 data did not include E1-E3 variables (3 missing data). Thus, we excluded the 3 data for analysis.

Out of obtained the 260 participants’ data, this study excluded missing data of the three perceived effects (E1, E2, E3), and this study used 257 valid data as a final sample data.

·        Page 9, line 216-217 – The beginning of the sentence should read “Of the participants…”

We change the parts as per your suggestion.

Page 9, line 219- qualify what you are saying – It should read, “The percentage of participants…

These %  indicate the percentage of participants of each category.

Of the participants, 49.03% had lived in their current living space for 20 years or more. Among the types of damage related to the GEJET, 45.33% of participants claimed damages from the tsunami, and 14.79% of participants indicated no damage. Regarding the preparation of meals, the percentage of participants cooking by themselves was 23.35%. The percentage of participants eating alone was 17.90%. Regarding health status, 36.19% of participants claimed poor SRH, and 3.5% claimed poor mental health (k6 ≥ 13).

·        Page 10, line 235-240 – the sentence is too long. Consider breaking into 2 or 3 so it is easy to understand.

The description just listed used variables, so it is hard to split the sentence.

·        Page 11, line 247 – Start the sentence with an article. It should read “The difference…”

We started the sentence as the difference・・・.

  1. Discussion

·        Are these findings consistent with any studies previously conducted or vice versa? It would be good to indicate this

We could not find the similar study in the previous study because cooking participation volunterr is really rare and rarely conducted in the world. This is unique and original intervention by the Ajinomoto Foundation. TAF conducted cooking support for stake holders and partners in developing countries. The cooking lecture was conducted just for stakeholders and di not include participants who just eat cooked meals.

TAF has not assessed the motivation and effects of the cooking supports. So, we believe this study would be the first study report.

·        Page 12, line 316 – There are extra characters in the sentence that form no word

We modified these parts as per your suggestions.

·        Page 12 line 317-328 – There seems to be some contradiction in what is being stated. In the preceding narrative, it looks like the classes yielded positive results for social cohesion/interaction, but in these lines the authors are saying volunteer supported cooking classes were low, no sitting together at dinning tables to eat etc. This piece needs to be presented more clearly.

The authors are saying volunteer supported cooking classes were low, no sitting together at dinning tables to eat etc.

>the sentences mentioned in the previous studies. No cooking class volunteer activities were conducted in the previous studies. These descriptions presented the current negative impact and problems in the disaster-affected areas. To tackle the issue, this TAF activities would be a useful solution to dissolve such issues.

We modified the parts as possible as your suggestions.

·        Page 13, line 361-367 – These sentences need to be reworked and made clearer.

·         

We revised these sentences as follows.

Previous studies focused on post-disaster food supply as a matter of physical survival in an emergency situation. By contrast, this study takes long-term post-disaster nutrition health to the next level, evaluating the balance of a diet and considering and prioritizing cultural dietary aspects that embrace traditional heritage and history. Such aspects encourage the reestablishment of socialization and fellowship around a table, as well as improvement in mental health and emotional recovery from the physical and emotional toll such crises can have.

Response to Reviewers’ Comments

Reviewer 2 Report

This study aimed to explore the impact of motivations of participants who had joined the cooking classes in the affected area of the Great East Japan Earthquake.

I felt that this manuscript is original and interesting.

However, it can be improved and I leave some comments as follows;

1. Authors designed their study in which dependent variables were increase in new acquaintances and friends, increase of outing opportunities, and gaining a new sense of purpose in life. I agree that these points are important. However, they seemed to be achieved even through other kind of events than cooking classes. Salon (tea ceremony), exercise classes, or any other meeting may substitute the role of cooking classes. I think that authors don't need to change dependent variables, however, they can stress the possibility of causing positive effect of cooking classes compared with other kind of meetings.

2. Line 50 "the ratio" can change into proportion

3. Line 103 the wording of "intervention" can change into other word because this study design was cross-sectional and the cooking class was not "intervention" designed in this study. 

4. L116 Please elaborate on raise awareness "of what."

5. Please describe additional information about the cooking class including how participants were recruited, how local people were informed on purpose of this class, criteria (if any) of participation, participation fee, frequency, presence of absence of classification of participants according to their sex, age, skills of cooking or other characteristics.

6.L177 Types of disaster damage can duplicate. In table 1, authors explained multiple answers were allowed. However, the number of total response was 257, supposing that every respondent answered that they suffered from just one damage. Did respondents really answered so?

7. Regarding tables, it is recommended that the tables are self-explanatory and should be able to understand independently. In table 2, additional information on what M1 through M6 indicated, and "a" through "o" meant each venue of cooking classes and what kind of independent variable were adjusted.

8. Table2: I couldn't understand authors' aim of setting the venue "a" as reference and compare with other venues.

9. Page 11. Lower part. No title was shown in figures. Please add a title and legend.

10. Discussion: All respondents of this study were participants of cooking class. However, large part of local residents were not participants and their intention about why they don't participate in the class could help implementing this activity in larger population. Authors can mention regarding the study sample is biased.

11. I do agree that this kind of cooking class is valuable and beneficial for local people. However, this class is operated by a company and not a self-governing society. When this support from outside of the community stop, will local people be able to maintain the achievement which set dependent variables in this study? Support from outside may have potential of inhibiting self-governing power of communities, I suppose.

12. As mentioned above, to gain effects of this study, possibility of combining a cooking event with other kind of activity because some parts of population are not interested in cooking and healthy behavior also include exercise habit. A previous study also assess the intervention of exercise class on mental health of survivors of the Great East Japan Earthquake and this study clarified that the program was effective in maintaining their subjective-wellbeing through meeting with neighbors.

Effectiveness of group exercise intervention on subjective well-being and health-related quality of life of older residents in restoration public housing after the great East Japan earthquake: A cluster randomized controlled trial. International Journal of Disaster Risk Reduction

https://doi.org/10.1016/j.ijdrr.2020.101630

Authors can elaborate their discussion on possibility of expanding target population by providing various kinds of classes. The above article may help adding this idea.

13. Limitation of this study may include absence of a control group of people participating in other events than the cooking class or people who don't attend any other event. 

14. Please check some typos. 

     L7: Grad was bold

    Table1: E2 missing 0 ⇒proportion should be also 0.

               E3 missing 0 ⇒proportion should be also 0.

    L125 : Data collection?

    L360: space between words is wider.

Author Response

Thank you very much for the thoughtful and constructive feedback you provided regarding our manuscript. We appreciate your taking the time to review our manuscript. Our responses to your comments are as follows:

1. Authors designed their study in which dependent variables were increase in new acquaintances and friends, increase of outing opportunities, and gaining a new sense of purpose in life. I agree that these points are important. However, they seemed to be achieved even through other kind of events than cooking classes. Salon (tea ceremony), exercise classes, or any other meeting may substitute the role of cooking classes. I think that authors don’t need to change dependent variables, however, they can stress the possibility of causing positive effect of cooking classes compared with other kind of meetings.

You may think the outcomes can be presented in other activities, but we would like to highlight the bys through cooking class participation. Social activities you mentioned are just as hobbies and leisure. Benefits bring only to the participants and the spillover is low to the whole community social cohesion. People who have the same aims tend to get together. People with the same interests tend to gather, and there is no diversity of their connections. Some older people with physical disabilities cannot enjoy those activities, although some participants can help them. But, the sense of accomplishment by oneself is low.

By contrast, eating is universal behavior and not a barrier to gather (salon, exercise, etc.). In this study, unbalanced eating habits are a problem in disaster-affected sites. TAF activity tackles the issue by improving dietary behavior and habit.

We focused on the eating behavior problem as well as social reconnection through TAF activity, not just a social leisure activity.

We added the strength the effects through cooking class, not other activities in the discussion section.

Given the study sites had problems of nutrition take (over salt intake ) and eating habits (eating alone), this study focused on not a leisure activity but food intake improvement with someone. Eating is a universal activity, and the participants are more diverse than other social activities. Unlike other activities, the TAF volunteer activity is a different approach to restoring social ties through activities aimed at improving diet. Eating together has the effects of achieving a sense of accomplishment through collaborative work in a short period of time, Moving hands by cooking make them feel independent, and stimulate to their five senses in cooking and eating a meal. During cooking, conversations naturally occur with people you meet for the first time or people you are not familiar with. We confirm that the secondary effects are large. The effects that social participation through cooking class is more suitable for improving nutrition and community cohesion than the activities in other previous studies in the post-disaster context.

As we demonstrated, the majority of the participants of this study were disaster victims(Tsunami or Fukushima Nuclear plant accident) Just after the disaster, they lived in evacuation centers for some months. Through the long evacuation center life,their nutritional status deteriorated  much. (see,Amagai T, Ichimaru S, Tai M, Ejiri Y, Muto A. Nutrition in the Great East Japan Earthquake Disaster.  Nutr Clin Pract. 2014 Oct;29(5):585-94. doi: 10.1177/0884533614543833.PMID: 25606634 ) .Then, we believed that holding cooking class might clearly contributed improved nutritional status of the disaster victims. In our study, participants also mentioned increased variety of food materials after participation of the cooking class. 

2. Line 50 “the ratio” can change into proportion

We modified the word, proportion.

3. Line 103 the wording of “intervention” can change into other word because this study design was cross-sectional and the cooking class was not “intervention” designed in this study.

We change the word, support.

4. L116 Please elaborate on raise awareness “of what.”

We added the explanation as follow:

raise awareness of social and nutritional issues.

5. Please describe additional information about the cooking class including how participants were recruited, how local people were informed on purpose of this class, criteria (if any) of participation, participation fee, frequency, presence of absence of classification of participants according to their sex, age, skills of cooking or other characteristics.

Some cooking classes were implemented as part of the Health Promotion Seminar held in every 2 months, and in total 6 times per one year, organized by local governments such as in Kesennuma City, Miyagi. For this Health Promotion seminar, participants paid 500 yen even for the cooking class. Participants, therefore, clearly understood the purpose of the cooking class.

We added the explanation as follow:

To recruit participants, the Ajinomoto Group and TAF approached local governments and public institutions (mainly social welfare councils), councils for improving eating habits, and temporary housing residents’ associations. The head of each organization announced the cooking classes to the residents on a community circular board (kairanban).

Porpose of this class was added in the following sentence.

The class was aimed at resolving social and nutritional issues in these communities after the disaster, through addressing cooking and eating behavior.

There is no critetia of participants and the characteristics. Everyone can join the class.

We do not have basic info about frequency, presence or absence of classification data.

But, we surveyed the questionnaire and got the respondence characteristics (frequency, presence of classification of participants), which can be found in the result section.

6.L177 Types of disaster damage can duplicate. In table 1, authors explained multiple answers were allowed. However, the number of total response was 257, supposing that every respondent answered that they suffered from just one damage. Did respondents really answered so?

Thank you for your suggestion. This is one selection. We deleted the note, multiple answers in table 1.

7. Regarding tables, it is recommended that the tables are self-explanatory and should be able to understand independently.

In table 2, additional information on what M1 through M6 indicated, and “a” through “o” meant each venue of cooking classes and what kind of independent variable were adjusted.

We modified more clearly the table 2 as per your suggestions.

8. Table2: I couldn’t understand authors’ aim of setting the venue “a” as reference and compare with other venues.

The venue ‘a’ is the lowest damage in the 15 venues by GEJET. We referenced the site.

9. Page 11. Lower part. No title was shown in figures. Please add a title and legend.

We presented the full title in figures.

Figure 3. Marginal effects of eating pattern on participation effects by motivation.

dy/dx is the discrete change in participation effects of margins by motivation, with eating pattern based on the fitted Poisson regression model in Table 2 (supplement A2).

10. Discussion: All respondents of this study were participants of cooking class. However, large part of local residents were not participants, and their intention about why they don’t participate in the class could help implementing this activity in larger population. Authors can mention regarding the study sample is biased.

As we explained, a majority of our cooking class was supported or organized by local governments affected by GEJET, or Fukushima Nuclear plant accidents. Therefore, the local governments equally, widely announced to the communities about our cooking class. However, the cooking class is normally held on weekdays, from 10:00 am to 13:00 pm. Therefore, the participants might have been limited; the aged people were more likely to participate. Moreover, male participants were shy and limited. 

Then, we organized the ‘Cooking class only for the males’ separately. The cooking class for the males held on Sundays. We did the best to invite more community people indeed.

As reviewers pointed out,  we invited all the community people, but the number of participants was limited affected by the cooking class’s time and location.

Our study did not have a control group, since we had a scope that comparing the effect of ‘no intervention group’ might be clearly low, or ‘no effect’. We decided to compare the different effects among sub-groups in the same intervention group might  be academically valuable.

As mentioned above, this study sample was biased, but we would like to emphasize that we invited all the residents equally.

We could not survey non-participants of the cooking class because we conducted the questionnaire survey only on participants. We added the limitation in the discussion section.

This study was conducted only with program participants, with no control group comprising people participating in other events, or people who had not attended any other event. A large group of local residents did not participate, and this study could not determine their reasoning for not participating in the class. Thus, the study sample is biased on controlled trials.

11. I do agree that this kind of cooking class is valuable and beneficial for local people. However, this class is operated by a company and not a self-governing society. When this support from outside of the community stop, will local people be able to maintain the achievement which set dependent variables in this study? Support from outside may have potential of inhibiting self-governing power of communities, I suppose.

The cooking class was prospected to semi-independent by local people and in the future plan, the activity would be conducted completely as an independent activity by local people.

At evacuation places and temporary housing, there are many people who receive support and support from all over the world and are often pampered and do nothing for themselves. In this project, TAF talked with their local partners and tried to help participants prepare, cook, and tidy up and not let them be “customers”.

Therefore, Ajinomoto Foundation addresses local councils and institutions to lead the activity be themselves by lecturing and teaching the methods for independence.

Actually, some local governments carry out independent activities by collecting participation fees from participants. We added this concern in the discussion section.

TAF promised to assist in facilitating independent operation of the class by local people in the future. In fact, a few years after TAF supports started, some local governments have carried out independent cooking class activities by collecting participation fees (about $ 4.5) from participants without TAF’s support.

12. As mentioned above, to gain effects of this study, possibility of combining a cooking event with other kind of activity because some parts of population are not interested in cooking and healthy behavior also include exercise habit. A previous study also assess the intervention of exercise class on mental health of survivors of the Great East Japan Earthquake and this study clarified that the program was effective in maintaining their subjective-wellbeing through meeting with neighbors.

Effectiveness of group exercise intervention on subjective well-being and health-related quality of life of older residents in restoration public housing after the great East Japan earthquake: A cluster randomized controlled trial. International Journal of Disaster Risk Reduction

https://doi.org/10.1016/j.ijdrr.2020.101630

Authors can elaborate their discussion on possibility of expanding target population by providing various kinds of classes. The above article may help adding this idea.

Some vunue conduct combined activity with cooking class. For example, a cooking class was included in one of the health promotion classes six per month.

Some cases included classroom lectures and gymnastics classes. However, we could not obtain data of how often and what kind of combination was performed in each venue in detail.

We would like to discuss the synergistic effect of the combination with other programs as the next study.

In this study, we could not conduct a randomized controlled trial because we recruited only participants of the cooking class.

This study has a potential unique effect that cannot be compared to the effects through previous activities and intervention reports.

13. Limitation of this study may include absence of a control group of people participating in other events than the cooking class or people who don’t attend any other event.

We added the limitation related to comment No.12 you mentioned.

This study was conducted only with program participants, with no control group comprising people participating in other events, or people who had not attended any other event. A large group of local residents did not participate, and this study could not determine their reasoning for not participating in the class. Thus, the study sample is biased on controlled trials.

14. Please check some typos.

     L7: Grad was bold

    Table1: E2 missing 0 ⇒proportion should be also 0.

               E3 missing 0 ⇒proportion should be also 0.

    L125 : Data collection?

    L360: space between words is wider.

We modified all the relevant parts that you mentioned.

14. Please check some typos.

    L7: Grad was bold

    Table1: E2 missing 0 ⇒proportion should be also 0.

    E3 missing 0 ⇒proportion should be also 0.

    L125 : Data collection?

    L360: space between words is wider.

We modified the relevant parts as per your suggestion.

Round 2

Reviewer 2 Report

Dear Authors,

Thank you for your detailed response to reviewers and revision of your manuscript.

Explanation in your response letter was excellent and you replied all questions from reviewers appropriately.